

# Providing choice of feedback affects perceived choice but does not affect performance

Gal Ziv[1], Ronnie Lidor[1] and Oron Levin[2,3]

[1] The Academic College at Wingate, Netanya, Israel
[2] KU Leuven, Leuven, Belgium
[3] Lithuanian Sports University, Kaunas, Lithuania

## ABSTRACT

**Background:** Autonomy or choice can lead to improved learning in various educational domains. The purpose of this online study was to examine whether giving participants a choice regarding the frequency of their received feedback (either after each individual trial or after a block of trials) in a computerized alternate task-switching task, will affect their performance.

**Methods:** Participants ($n$ = 148) were randomly assigned to three groups: choice group ($n$ = 49), online feedback group ($n$ = 51), and summary feedback group ($n$ = 48). From those three groups we created two groups: a choice group and a no-choice group ($n$ = 49 in each group). All participants performed eight familiarization trials, a pre-test of 24 trials, five blocks of 24 trials for practice, and a post-test of 24 trials. After completing the task, the participants were asked about their perceived feeling of choice and completed the short form of the International Positive and Negative Affect Schedule.

**Results:** The participants in the choice group had higher perceived choice compared with the participants in the no-choice group (8.41 $vs$ 5.47 out of 10, respectively). However, this higher perceived choice did not materialize into better performance during practice or in the post-test.

## INTRODUCTION

Autonomy (or choice) is considered a basic biological (*Leotti, Iyengar & Ochsner, 2010*) and psychological (*Deci & Ryan, 2008*) need. Providing individuals with choice enhances their intrinsic motivation (*Patall, Cooper & Robinson, 2008*) and self-efficacy (*Wulf, Chiviacowsky & Cardozo, 2014*). Self-efficacy, in turn, is a predictor of performance (*Feltz, Chow & Hepler, 2008*). Autonomy support can enhance motor performance and learning (*Wulf & Lewthwaite, 2016*), enhance learning engagement (*Jang, Reeve & Deci, 2010*), and lead to more adaptive learning behaviors (*Schweder & Raufelder, 2021*). Autonomy can be offered by allowing participants to choose when to receive feedback (*i.e.*, self-controlled feedback, *Chiviacowsky, 2014*); by providing choice of training devices, *Hartman (2007)*; by using autonomy-supportive instructional language, *Hooyman, Wulf & Lewthwaite (2014)*; by letting participants choose the amount of practice, *Lessa &*

Corresponding author
Gal Ziv, galziv@wincol.ac.il

*Chiviacowsky (2015)*; or even by providing choices that are irrelevant to the task at hand (*e.g.*, choose the color of a golf ball, *Lewthwaite et al., 2015*).

One instructional method that is often used to improve performance and learning is augmented feedback. Feedback can be given in various forms that are either controlled by the instructor or by the learner. For example, learners can be given choice regarding the type, the timing, and the frequency of the feedback they receive. Feedback, for example, can be given after each trial (immediate or online feedback) or at the end of a series of trial (summary feedback). *Guadagnoli & Lee (2004)* suggested that whether online feedback or summary feedback yields larger learning effects depends on task difficulty. Specifically, these authors argued that for a difficult task, more frequent feedback might yield larger learning effects compared with summary feedback, while the opposite is true for a simpler task.

This moderating effect of task difficulty on the benefits of feedback frequencies has been shown by *Guadagnoli, Dornier & Tandy (1996*; Exp. 2). In this experiment, participants tried to strike a padded force transducer with their right fist to match a predefined force (easy task) and with their right fist and immediately followed by their left fist (complex task). Findings of this study showed that for the simple task, providing summary feedback after 15 trials led to better performance in a retention test compared with feedback after five trials or after each individual trial. This was true for both novice and experienced participants. In contrast, for the complex task, providing novice participants with feedback after each individual trial led to better retention compared with the provision of summary feedback after 15 trials.

In light of the abovementioned findings, if feedback frequency has a different impact on performance according to task difficulty, then it could be important to examine whether providing participants with the autonomy to choose their feedback frequency would lead to improved performance, even if, for example, they choose the "*wrong*" feedback (*e.g.*, summary feedback for a difficult task). In the current study we implemented a computerized alternate task-switching task that includes more difficult switch trials compared with less difficult no-switch trials (*Monsell, 2003*).

Therefore, the purpose of the current study was to examine whether providing participants with a choice of receiving feedback after each trial or receiving only summary feedback at the end of a block of trials would lead to improved performance in an alternate task-switching task. We hypothesized that: (a) participants in the choice group would outperform participants in the no-choice group during practice and would show greater improvement from pre- to post-test, (b) participants in the choice group would experience greater positive affect (as observed in previous studies, *e.g.*, *Joussemet et al., 2004*; *Lemos et al., 2017*) compared with participants in the no-choice group, (c) participants in the online feedback group would perform better than participants in the summary feedback group (because such knowledge of results can serve as guidance during practice and lead to improved performance; *Salmoni, Schmidt & Walter, 1984*), and (d) if one type of feedback would be better than another, we would observe preference-performance dissociation among some participants (*e.g.*, participants will choose a type of feedback that will be detrimental to their performance). Indeed, this dissociation has been reported previously

in a number of tasks: (a) a flying task in novices (*Wright & O'Hare, 2015*), (b) choice-RT tasks (*Ziv & Lidor, 2021*), and (c) a golf-putting task (*Ziv et al., 2020*). In addition, a manipulation check will show that participants in the choice group will perceive greater choice than participants who received no choice.

## MATERIALS AND METHODS

### Experimental approach

The study was designed and conducted on a cloud-based experimental software (www.gorilla.sc; *Anwyl-Irvine et al., 2020*). We aimed at examining whether allowing participants to choose the feedback frequency they receive – feedback after each trial or summary feedback after a block of 24 trials, would lead to improved performance (*i.e.*, faster reaction time (RT) and higher accuracy) in an alternate task-switching task compared with participants who received feedback without the ability to choose.

To answer this question in a laboratory-based experiment, we would have to allocate the participants to two groups: a choice group and a no-choice group. For every participant who chooses the type of feedback in the choice group, the same feedback is given without choice to one of the participants in the no-choice group. However, in an online study this is not always possible.

Therefore, participants were first allocated automatically into three experimental groups: (a) a summary feedback group – feedback after each block of trials, (b) an online feedback group – feedback after each trial (thumbs up or down), and (c) a choice group – a choice of either summary or online feedback. These three groups allowed us to create the two required groups: (a) a choice group (which was the original group) and a no-choice group (equal in numbers to the choice group and based on participants from the original online feedback and summary feedback groups).

### Pre-registration and raw data repository

The study was pre-registered on aspredicted.org (https://aspredicted.org/d4gv3.pdf). Analyses that were not pre-registered are reported in the "exploratory analysis" section. The raw dataset is available on OSF (https://osf.io/p5wgx/?view_only=0ca32fa378874982b6ca28df88604413).

### Participants

We recruited participants on Prolific (www.prolific.co), an online platform that allows individuals with access to the Internet to participate in online studies. The participants were paid 2.5 British Pounds for their participation. The participants were between the ages of 18–35, were fluent in English, and reported no chronic illnesses.

We used G * Power (*Faul et al., 2007*) to perform *a priori* power analysis for a two-way analysis of variance [ANOVA; group (choice/no-choice) × Test (pre/post)] comparing RTs. The effect size for this analysis was based on a recent study that examined the provision of choice in similar online tasks (*Ziv & Lidor, 2021*). This estimation showed an

expected effect size of $\eta^2_p = 0.13$ corresponding to Cohen's f = 0.38 suggesting that participants in the choice group will show faster RTs compared to the participants in the no-choice group. Because small studies tend to have larger effect sizes (the Winner's Curse; *Button et al., 2013*), we decided to reduce the effect size by 20% and used Cohen's f = 0.30. We entered this effect size into the power analysis with the following parameters: alpha (two-sided) = 0.05, power = 0.90, and correlation among repeated measures = 0.5. The power analysis suggested that 90 participants (45 for the choice group and 45 for the no-choice group) were required to detect differences between groups with 85% power.

To account for possible dropouts, we started with a total of 151 volunteers who completed the experiment. Based on our pre-registration exclusion criteria for participation, three participants were removed from the study. This left us with 148 participants who were randomly assigned by the software to three groups: (a) choice group (*n* = 49, 36 females, mean age = 25.4 ± 4.4 years); (b) online feedback group (*n* = 51, 34 females, mean age = 27.2 ± 4.7 years); and (c) summary feedback group (*n* = 48, 29 females, mean age = 26.8 ± 4.8 years). From those three groups, we then created two groups: choice group (*n* = 49, 24 chose to receive summary feedback and 25 chose to receive online feedback), and no-choice group (*n* = 49, the first 24 participants from the summary feedback group and the first 25 participants from the online feedback group).

All participants completed an electronic informed consent form on the study's website prior to participation. If participants chose not to participate, the software forwarded them to a thank you screen, and they were excused from the study. The study was approved by the Ethics Committee of the Academic College at Wingate (approval # 308).

## Task

In this task, a blue or green rectangle or square were shown at the upper part or the lower part of the computer screen. When the shape was located at the upper part of the screen, the participant was required to press "f" if the shape was blue and to press "j" if the shape was green. However, when the shape was located at the bottom part of the screen, the participant was required to press "f" if the shape was square, and "j" if it was rectangle. The stimuli were presented with no time limit until the participants pressed a key. The presentation of the shapes at the upper or lower parts of the screen alternated every two shapes.

## Procedure

Data collection was done remotely as each participant participated on their own computer. After reading the informed consent form and agreeing to participate in the study, all participants performed eight familiarization trials of the task. In these trials, both online feedback (a thumbs-up or thumbs-down image appeared after each trial) and summary feedback (number and percentage of correct trials) were provided. Then, the participants performed a pre-test of 24 trials with no feedback. This was followed by a practice session that included five blocks of 24 trials, where feedback was provided based on group

affiliation. The participants in the online feedback group saw a thumbs-up symbol or a thumbs-down symbol for a correct or an incorrect key press, respectively, after each trial but did not receive summary feedback. The participants in the summary feedback group, on the other hand, only received information about the number of correct responses and percentage of correct responses after each of the five blocks of 24 trials (*e.g., "You scored 21 out of 24. That's 88% correct"*) but did not receive online feedback. The participants in the choice group were asked whether they wanted to receive online or summary feedback and received the appropriate type of feedback based on their choice. After the five practice blocks were completed, the participants performed a post-test of 24 trials with no feedback. Finally, participants were asked to rate whether they believed that they had a choice over the type of feedback they received on a scale of 1 (not at all) to 10 (very much), and were then asked to complete the short form of the International Positive and Negative Affect Schedule (PANAS). This questionnaire was selected because it was developed in order to provide valid assessment of affect cross-culturally, and therefore is suitable for an online study where participants are from different countries and/or cultures (*Thompson, 2007*). The questionnaire is composed of 10 items that are rated on a scale of 1 (not at all) to 5 (very much): Determined, attentive, alert, inspired, active, afraid, nervous, upset, ashamed, and hostile. After completing this questionnaire, the participants were thanked for their participation and the study ended.

## Data analyses

In our pre-registration, we decided to exclude mean RTs of under 300 ms or over 2,500 ms. This happened five times in the pre-test and seven times during practice. We also decided to exclude blocks with 13 or less correct responses out of 24. This occurred nine times in the pre-test, three times during the practice stage, and once in the post-test. Finally, we excluded participants who had at least two blocks during practice with 13 or less correct responses. This led to the exclusion of three participants from the study.

Normality was assessed using Kurtosis and Skewness values (values <2 were considered acceptable). For RT, this was mostly the case (except for Kurtosis values of 4.06, 3.73, and 3.24 for Blocks 3, 4, and 5, respectively, during practice), and therefore we used parametric statistics to analyze these data. For the number of correct responses, this was not the case (except for the pre-test) and therefore non-parametric statistics were used. We conducted a two-way ANOVA (Group (choice/no-choice) × Test (pre/post)) to assess differences in RTs. In addition, we conducted a two-way ANOVA (Group (choice/no-choice) × Block (1–5)) to assess performance during practice, and a three-way ANOVA (Group (online feedback/summary feedback) × Switch (yes/no) × Test (pre/post)) to assess differences in the switch or the no-switch trials only. Bonferroni post-hoc analyses and 95% confidence intervals were used for post-hoc testing. For analyzing the correct responses we used the Mann-Whitney test. Analyses were conducted using SPSS (version 25; IBM, Armonk, NY, USA) and the significance level was set at alpha = 0.05.

## RESULTS

There were no gender differences in any of the dependent variables.

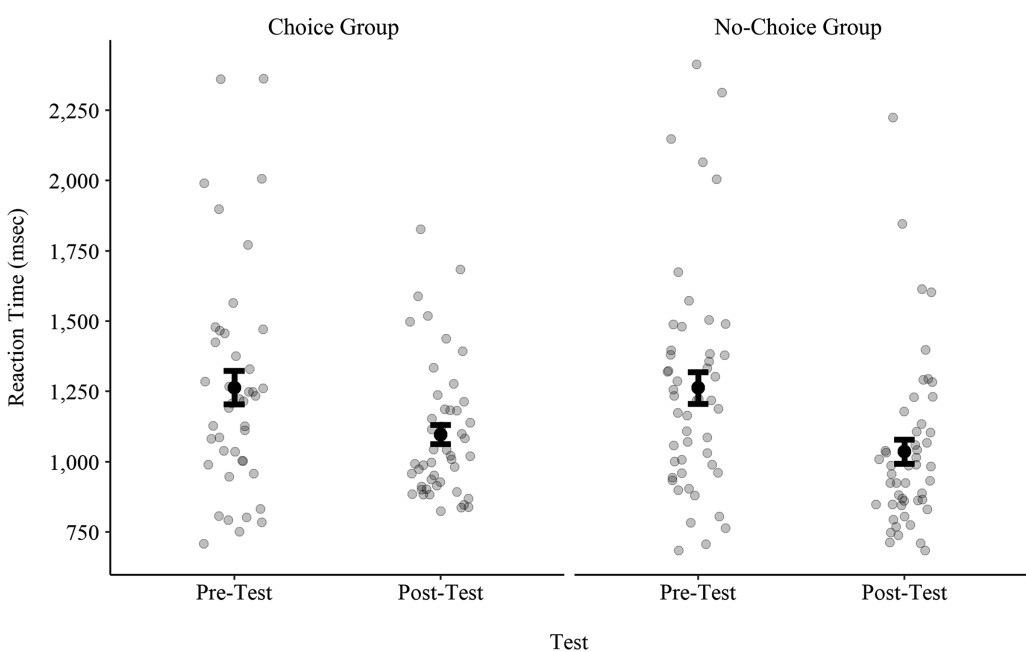

**Figure 1 Reaction times in the pre-test and in the post-test for both the choice and no-choice groups.**
Black dots and error bars represent the mean and the standard error, respectively. Light grey circles
represent values of individual participants.

## Manipulation check - perception of choice

The participants in the choice group rated their perception of choice on a scale of 1 to 10
higher (8.41 ± 2.42) than the participants in the no-choice group (5.47 ± 3.37) during the
experiment, $t(96) = 4.96$, $p < 0.001$, Cohen's d = 1.00.

## Differences between pre- to post-test
### RT

RTs for the pre-test and the post-test for both choice and no-choice groups are presented
in Fig. 1. A two-way ANOVA (Group (choice/no-choice) × Test (pre/post)) with repeated
measures on the Test factor revealed a Test effect, $F(1, 90) = 32.45$, $p < 0.001$, $\eta^2_p = 0.27$.
The RT in the pre-test (1,265.54 ± 395.00 ms) was longer than the RT in the post-test
(1,047.42 ± 259.22 ms). There was no Group effect, $F(1, 90) = 0.06$, $p = 0.81$, $\eta^2_p = 0.00$, and
no interaction, $F(1, 90) = 0.06$, $p = 0.81$, $\eta^2_p = 0.00$.

### Correct responses

A Mann-Whitney test revealed no differences between the choice group and the no-choice
group in the pre-test ($U = 1,157.00$, $p = 0.53$; 10.88 ± 1.50 *vs* 10.93 ± 1.52, respectively) or in
the post-test ($U = 1,198.00$, $p = 0.86$; 11.64 ± 0.64 *vs* 11.65 ± 0.63, respectively).

## Differences between practice blocks
### RT

A two-way ANOVA (Group (choice/no-choice) × Test (pre/post)) with repeated measures
on the Block factor revealed no Group effect, $F(1, 93) = 0.44$, $p = 0.51$, $\eta^2_p = 0.01$, no Block

effect, $F(3.05, 284.01) = 2.01$, $p = 0.11$, $\eta^2_p = 0.02$, and no interaction, $F(4, 372) = 0.30$, $p = 0.88$, $\eta^2_p = 0.00$.

### Correct responses

A Mann-Whitney test revealed no differences between the choice group and the no-choice group in any of the practice blocks ($p$ values > 0.26 for Blocks 1–4, $p = 0.045$ for Block 5 – not significant after using the false discovery rate method to account for multiple comparisons, *Benjamini & Hochberg, 1995*). The mean number of correct responses ranged between 22.31 and 23.04 out of 24 in all five blocks.

## Differences between types of feedback

### RT

A three-way ANOVA (Group (online feedback/summary feedback) × Switch (yes/no) X Test (pre/post)) revealed a Switch effect, $F(1, 89) = 171.14$, $p < 0.001$, $\eta^2_p = 0.66$. The RTs for switch trials (1,356.95 ± 341.18 ms) were longer than the RTs for no-switch trials (1,031.76 ± 254.51 ms). There was also a Test effect, $F(1, 89) = 59.97$, $p < 0.001$, $\eta^2_p = 0.40$. RTs in the pre-test (1,327.30 ± 364.30 ms) were longer than the RTs in the post-test (1,104.61 ± 314.12 ms). There was no Group effect, $F(1, 89) = 0.08$, $p = 0.78$, $\eta^2_p = 0.00$, no Group × Test interaction, $F(1, 89) = 2.52$, $p = 0.12$, $\eta^2_p = 0.03$, no Group × Switch interaction, $F(1, 89) = 0.15$, $p = .70$, $\eta^2_p = 0.00$, no Switch × Test interaction, $F(1, 89) = 1.47$, $p = 0.23$, $\eta^2_p = 0.02$, and no Group × Switch × Test interaction, $F(1, 89) = 0.05$, $p = 0.82$, $\eta^2_p = 0.00$.

### Correct responses

There were no differences in correct responses between the participants in the online feedback group and the participants in the summary feedback group in the pre-test or the post-test (all $p$ values > 0.13).

## PANAS scores

There were no differences between the choice group and the no-choice group in any of the 10 attributes in the PANAS questionnaire (all $p$ values > 0.09). There were also no differences between these attributes when examining all three original groups (*i.e.*, choice, online feedback, and summary feedback; all $p$ values > 0.4).

## Exploratory analysis

### Variability of RT

As seen in Fig. 1, the variability in the RTs of the choice group in the post-test appear to be smaller than the variability in all other conditions. Indeed, the standard deviation of the choice group, which was 399.05 ms in the pre-test, dropped to 207.39 ms in the post-test. In the no-choice group, the standard deviation, which was 395.53 ms in the pre-test, dropped to 299.15 ms in the post-test.

### Post-test analysis with the pre-test as a covariate

It has been previously suggested that a one-way ANCOVA on a post-test with a pre-test as a covariate provides more statistical power and statistical precision compared with a

two-way ANOVA (Group × Test) with repeated measures on the Test factor (see *Rausch, Maxwell & Kelley, 2003*; *Senn, 2006*). However, similar to our main analyses, this ANCOVA did not reveal any significant differences between the choice group and the no-choice group, $F(1, 89) = 0.206$, $p = 0.65$, $\eta^2_p = 0.00$. In addition, no significant differences were found when examining the original three groups, $F(2, 130) = 0.85$, $p = 0.43$, $\eta^2_p = 0.13$.

## DISCUSSION

The purpose of the current study was to examine whether providing participants with a choice over feedback delivery would lead to improved performance in an alternate task-switching task. We hypothesized that choice would lead to improved performance, affect, and perceived choice. In addition, we hypothesized that online feedback would lead to better performance compared with summary feedback. The only hypothesis that was supported by the data of the current study was that providing participants with a choice leads to higher perceived choice. There were no other differences in performance or in positive affect between groups.

Our findings do not support previous findings regarding autonomy support (*e.g.*, *Chiviacowsky, 2014*; *Iwatsuki et al., 2017*; *Iwatsuki & Otten, 2020*; *Lewthwaite et al., 2015*). For example, one previous study provided choice regarding the order of performance of two computerized RT-based tasks (a choice-RT task and a Simon task) (*Ziv & Lidor, 2021*). Participants who were given a choice had faster RTs compared with participants who were not given a choice.

While it is not directly apparent why providing choice of feedback did not lead to improved performance in the current study, a number of explanations can be discussed. First, it is possible that the task was too easy to reveal any differences. However, we do not think this is the case because similar and even easier computerized tasks (*e.g.*, a Simon task) have been used in previous studies and were sensitive enough to show differences in performance between a choice group and a no-choice group (*e.g.*, *Ziv & Lidor, 2021*). It is also possible that the task may have been too difficult, thereby leading to relatively mild improvements. In such a case, a lengthier intervention that includes a larger number of practice blocks, as well as a retention test that takes place 24–48 h after the practice session could expose the benefits of an autonomy support intervention. In short interventions, the effect of autonomy support may simply not be robust enough. Indeed, differences in retention or transfer tests, but not during practice are frequently found in studies on various motor learning interventions (*e.g.*, *Ávila et al., 2012*; *Chiviacowsky & Lessa, 2017*; *Ziv, Ochayon & Lidor, 2019*).

Another possible explanation for the null findings in the current study is that our post-test took place immediately after practice (*i.e.*, immediate retention) and not, for example, after a longer period (*e.g.*, 24–48 h; delayed retention). Indeed, the effects of practice are often not observed in an immediate-retention test, however may materialize in a delayed-retention test. *Kantak & Winstein (2012)*, in their review, showed that in 26 out of 41 studies (63%), the effects of practice on performance in an immediate-retention test and a delayed-retention test differed. Out of these 26 studies, in 19 studies (73%)

significant effects of practice on performance were found in the delayed-retention test but not in the immediate-retention test, and only in three studies the effects were seen in the immediate-retention test but not in the delayed-retention test.

A possible explanation for the lack of effect in an immediate-retention test is that the memory consolidation process takes place in the hours after learning and during sleep, and therefore it has no time to materialize in an immediate-retention test (*Kantak & Winstein, 2012*). In a more recent meta-analysis (*McKay et al., 2022*), reduced feedback frequency did not lead to any changes in either immediate- or delayed-retention tests. The lack of differences in performance between the summary feedback group and the online feedback group supports the findings of McKay et al.'s meta-analysis.

While most published studies on autonomy support found it to be beneficial for motor performance and learning, some studies did not find this effect (*e.g.*, *McKay & Ste-Marie, 2020a*, *2020b*; *Yantha, McKay & Ste-Marie, 2021*). *McKay & Ste-Marie (2020a)*, for example, found trivial and statistically insignificant differences between participants who were given a choice over the color of a golf ball that they used compared to no-choice participants who were not given this choice. *McKay & Ste-Marie (2020b)*, revealed no differences in dart-throwing performances between participants who were given a choice of dart color compared with participants who were not given this choice. Similar to the results of the current study, this lack of difference in performance was accompanied by significantly greater perceived choice in the choice group. *Yantha, McKay & Ste-Marie (2021)* showed that self-controlled feedback schedules are not better than no-choice or predetermined feedback schedules when learning a golf-putting task. Finally, a recent meta-analysis on the effects on self-controlled learning (*McKay et al., 2021*) suggests that when considering both published and unpublished literature, the benefits of self-controlled practice on learning are very small (Hedge's $g = 0.11$; compared to 0.44 for published studies only). This meta-analysis suggests that bias inflates the effect size of autonomy interventions. The results of the current study support the findings of this meta-analysis and the abovementioned three studies. However, we found smaller variance in the RTs during the post-test in the choice group, compared with the no-choice group. While this finding is only descriptive, it may suggest that the autonomy intervention provided some benefit at least for some participants in the choice group. It is possible, for example, that providing choice of feedback leads to more attentiveness towards the received feedback, which in turn leads to more attention to the task, thus leading to more consistency in performance. This differences in variance can be further examined in additional studies.

Regardless of choice, based on Guadagnoli and Lee's challenge point framework (2004), we hypothesized that the task performed in our study would be difficult for inexperienced participants, and thus online feedback would lead to better performance compared with summary feedback. This hypothesis was not supported by the data. Therefore, it is possible that this task was not difficult enough and that with a more complex, perhaps real-world task (*e.g.*, a serve in tennis, a golf-putting task), differences between feedback frequencies will materialize. However, as *McKay & Ste-Marie (2020a)* showed, different feedback schedules even in a golf-putting task may only lead to trivial and statistically insignificant differences in performance. Finally, the findings of *Guadagnoli, Dornier &*

*Tandy (1996)* indicate that differences in performance following different feedback frequency schedules were only seen in a retention test. In the current study, such a test was not included as we only measured performance.

## Study limitations

This study has a number of limitations. First, we calculated our sample size based on a previous study that used the same tasks (*e.g.*, *Ziv & Lidor, 2021*), and to be cautious reduced the effect size by 20%. This led us to a Cohen's d of 0.6. However, a recent meta-analysis on autonomy support found Hedge's g of 0.54 for the published literature on self-controlled learning (see *McKay et al., 2021*). It is then possible that our sample size was not large enough to detect this smaller effect size.

Second, in such an online study, it is possible that at least some of the participants did not pay attention to the task. However, the mean RTs in the current study (approximately between 1,000 to 1,400 ms) are similar to mean RTs reported in similar tasks performed under laboratory conditions. *Salthouse et al. (2000)*, for example, reported values of over 1,000 ms in a switch trail making test. *Minear & Shah (2008)* reported RTs for three types of switching tasks. For one task, in the first block of practice, RTs were 889 and 1,027 ms for the no-switch and switch trials, respectively, and for the last block of practice these values were 697 and 763 ms. For a second task, in the first block of practice, RTs were 926 and 1,333 ms for the no-switch and switch trials, respectively, and for the last block of practice these values were 714 and 973 ms. For a third task, in the first block of practice, RTs were 1,020 and 1,395 ms for the no-switch and switch trials, respectively, and for the last block of practice these values were 740 and 1,011 ms.

In addition, as pre-registered, we removed from the study the RTs of participants who had less than 13 correct responses in a block of 24 trials because such results suggest that the participant was not attentive to the task. In future studies, researchers can use manipulation checks (see, for example, *Oppenheimer, Meyvis & Davidenko, 2009*). Another possible solution for participants' attentiveness to the task in online studies is using a within-group design. Such a design can shed more light on each individual participant's change in performance, and therefore allow more control over varying motivation and attentiveness between participants. It should be noted though that studies that compare performance in computerized tasks between laboratory and remote participation can be useful. A few studies have reported such comparisons for various aspects of online studies and their results mostly suggest that online studies can provide reliable data (*e.g.*, *Dandurand, Shultz & Onishi, 2008*; *Gould et al., 2015*). More specifically, *Crump, McDonnell & Gureckis (2013)* showed similar results of tasks such as Stroop, task-switching costs, the Flanker test, and the Simon test when performed in the laboratory or online.

One final limitation is missing a control condition of a simpler RT task that requires the use of the same two keyboard keys (*e.g.*, a choice-RT task). The differences between the RTs of such a control task and our main switching task could have shed more light on the difficulty and the level of attention required for this task. However, one previous online study required participants to perform a choice-RT task and a Simon task (a choice-RT

task that requires inhibition, and therefore is more difficult to perform) with the same two keyboard keys used in the current study and showed the expected differences in RTs in both tasks (choice RT < Simon RT) (*Ziv & Lidor, 2021*). In addition, *Crump, McDonnell & Gureckis (2013)* study showed that task-switching costs could be found in online studies, similarly to laboratory-based studies. Therefore, we believe that the observations in the current online study represent an accurate assessment of participants' performance.

## CONCLUSIONS

In summary, the results of the current study suggest that providing participants with a choice of feedback increase perceived choice, but it does not affect performance when completing a computerized alternate task-switching task. We suggest that the effects of autonomy support on performance and learning are not straightforward and may be moderated by various variables or conditions.

### Funding
The authors received no funding for this work.

### Competing Interests
The authors declare that they have no competing interests.

### Author Contributions
- Gal Ziv conceived and designed the experiments, performed the experiments, analyzed the data, prepared figures and/or tables, authored or reviewed drafts of the article, and approved the final draft.
- Ronnie Lidor conceived and designed the experiments, authored or reviewed drafts of the article, and approved the final draft.
- Oron Levin conceived and designed the experiments, authored or reviewed drafts of the article, and approved the final draft.

### Human Ethics
The following information was supplied relating to ethical approvals (*i.e.*, approving body and any reference numbers):

This study was approved by the ethics committee of the Academic College at Wingate (Approval # 308).

### Data Availability
The raw dataset is available at OSF: Ziv, Gal. 2022. "Choosing Feedback Online Study." OSF. June 7. osf.io/p5wgx.

### Supplemental Information
Supplemental information for this article can be found online at http://dx.doi.org/10.7717/peerj.13631#supplemental-information.

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
