# Peer review of "Providing choice of feedback affects perceived choice but does not affect performance"

_PeerJ, doi:10.7717/peerj.13631_

## Round 0.1 · original submission · Major Revisions

I agree with reviewers' comments reported below; please, accurately address them to further improve the overall quality and clarity of the manuscript.

Reviewer 1 ·

Basic reporting

The authors provide a well-written and concise manuscript describing an experiment that tested the OPTIMAL theory hypothesis that autonomy would benefit learning and performance. Additional hypotheses pertaining to positive affect (greater with autonomy support), feedback (performance greater with 100% feedback frequency), and feedback preference-performance dissociation (some learners would prefer a feedback schedule that was less effective).
The authors should be commended for a) preregistering their study, b) providing the raw data from their experiment, and c) conducting an a priori power calculation in an effort to adequately power their experiment. While I do have a number of comments/concerns that may need addressing before publication, I do believe this paper provides a contribution to the literature.

1. Basic Reporting

The manuscript provides a clear description of the experimental design and results. I was easily able to evaluate the methods in this paper and the authors should be commended for this.

My comments:

1. Perhaps the most important concern I have with respect to the rationale for this experiment is the use of reaction time task to test predictions from motor learning theory. The task required perceptual decision making, not motor skill. Indeed, with almost any motor skill there would be a highly significant block effect during acquisition (especially with a sample of this size). In this experiment there was no evidence of improvement across blocks, although there was a difference between pre- and post-tests suggesting some improvement did occur.

While I think that testing the impact of choice and feedback schedule on a perceptual decision-making task is worthy of investigation – real world motor behavior often involves rapid decision making, after all – I do think the authors should acknowledge that they are not testing the motor learning and performance predictions forwarded in OPTIMAL theory. Indeed, Wulf and Lewthwaite write in their theory (2016, pg. 1383):

“The foundation of our theory is motor behavior. That is, we are less concerned with skills that have a strong cognitive or decision-making component (i.e., learning “what”; see Shmuelof & Krakauer, 2011). Rather, of concern is primarily the learning of “how” to bring about the coordinated or skilled control of (complex) movement, for which the quality of movement execution is of primary importance (Shmuelof & Krakauer, 2011).”


2. It is unclear on what basis the authors form their predictions that positive affect will be influenced by autonomy support.
3. It is also unclear why the authors predicted a dissociation between chosen feedback schedule and the most effective schedule.
4. I assume that the authors predicted online (or 100%) feedback would be more effective than summary based on the guidance hypothesis (Salmoni et al., 1984), but I don’t see reference to that or an alternate rationale.
5. The authors should expand on their discussion that the that their post-test is an immediate retention test and discuss the potential implications of this measurement choice compared to a delayed retention test. See for example Kantak & Winstein, 2012 but also McKay et al., 2022 for discussions.

Experimental design

Experimental Design

The authors designed an experiment that appropriately addressed their research questions. I have minor quibbles that may be useful for the authors to consider, either in this paper or future papers.

1. Power analysis: I appreciate that the authors conducted an a priori power analysis to determine the sample size for this study. However, I have a couple points to consider. First, as the authors note in their discussion, there was a meta-analysis of self-controlled practice research available. Instead of using the estimates from that meta-analysis, the authors relied on a single estimate from a previous study that was obviously too large to be the true population parameter. To their credit, the authors reduced the estimate by 20%, but even still they powered for an effect of f = .30 or d = .60. Even the published experiments in the meta-analysis by McKay et al. (2021) only estimated an effect of d = .54, so the estimate in this power analysis was a bit optimistic.
2. Power analysis 2: As a technical follow up, I noted that the authors use partial eta-squared from a previous study to calculate the effect size in G*Power. Please be aware that the default settings in G*Power are not the same as in SPSS when it comes to partial eta-squared. As such, unless the settings are changed in G*Power to “as in SPSS,” the resultant sample size calculations in G*Power are substantially under-estimated.
3. My points above are meant only to facilitate a more accurate estimate of the a priori power the present study had to detect a range of effects. Of course, we do not know the true effect so we cannot know the actual power of the study.
4. Analyses: First, while I can understand why the authors chose to create a yoked-group from their two feedback groups, I think overall this strategy was quite inefficient. Why not compare the autonomy group to both groups combined, for example? The unequal sample sizes can be handled a variety of ways. Also, why not compare the three groups and follow up with pairwise comparisons? I think either method would have had some benefits compared to the approach the authors landed on – though I should be clear I do not think the authors’ approach was invalid.
5. Similarly, while not invalid, the mixed ANOVA approach to pre-post data is not ideal and ANCOVA is the preferred method (see Senn, 2006).
6. As a final note, I appreciate the analyses were preregistered and I support the authors sticking with their originally planned analyses as their primary results. If the authors choose to take my comments above into account, they may consider adding them as supplemental results or exploratory analyses.

Validity of the findings

The findings reported in this manuscript appear valid. The discussion of those findings with respect to OPTIMAL theory, while warranted, need to be better contextualized to acknowledge the difference between the task used in the present study and the type of movements OPTIMAL theory makes predictions about. Most previous research on autonomy and motor performance has focused on motor skills rather than decision making, so the authors should help the reader understand the gap the present research fills.

Additional comments

I appreciate the opportunity to review this paper.

References:

Kantak, S. S., & Winstein, C. J. (2012). Learning–performance distinction and memory processes for motor skills: A focused review and perspective. Behavioural brain research, 228(1), 219-231.
McKay, B., Hussien, J., Vinh, M. A., Mir-Orefice, A., Brooks, H., & Ste-Marie, D. M. (2022). Meta-analysis of the reduced relative feedback frequency effect on motor learning and performance. Psychology of Sport and Exercise, 102165.
Senn, S. (2006). Change from baseline and analysis of covariance revisited. Statistics in medicine, 25(24), 4334-4344.

Reviewer 2 ·

Basic reporting

This is a well-powered study that was conducted online. Pre-registration and data transparency are especially appreciated. The authors found that autonomy support by providing choice of feedback during performance leads to enhanced agency or autonomy but did not result in improvements in performance or affect. The use of remote online testing is also appreciated. While relevant and useful for remote human subject testing under current (pandemic) difficulties of in-person testing, it does have important drawbacks (see next section). The most important issue is related to establishing appropriate controls (see next section).

Other important issues concern the definition of certain terms and methodological description are listed here:

First, the use of the term “yoked,” to describe the two feedback groups is inaccurate. Yoking does not simply mean that the same number of participants be in the autonomy and feedback groups, but rather the order and manner of stimulus presentation. Any assignment to group that is performed ex post facto (as was done in this study) is non-experimental and should not be characterized using the same terminology as from an experimental paradigm. Therefore, this reviewer suggests that the feedback groups be described as they are and if they were combined for analysis, that that be described appropriately as well. It is noted that this is a drawback of online testing, however, conflating the feedback groups with a traditional yoked group is misleading (e.g., Line 100). Once again, this raises the larger issue of setting appropriate controls in this experiment, and it is my opinion that for the online mode of testing, a within-group control (by setting control/baseline conditions) will be more fruitful and convincing rather than a between-group control (however, there may be other clever ways of setting up between-group control). In this regard, the pre-post design used in this study is useful, but falls short for purposes of determining the effect of the task itself.

Second, the result that self-reported perceived choice was enhanced in the autonomy support group should be clearly distinguished as a check of experimental manipulation and not a hypothesized effect. In other words, that individuals in the autonomy perceived greater choice is (and should be) an expected effect; if this weren’t the case, then there wouldn’t by definition be an experimental group.

Third, there is no clear rationale in the Introduction for why the authors hypothesized a change in affect. By what means might autonomy support enhance affect? Appropriate literature must be cited, and the use of PANAS should be justified.

Lastly, although the raw data are provided, it seems that it was filtered for outliers. For full transparency, please provide data with outliers. Critically, there is no mention outlier detection or their treatment in the paper It is acceptable to perform outlier detection and removal, but a clear a comprehensive explanation for the reason and method for doing so should be described in the Methods section.

Experimental design

One important consideration for online tests across disciplines, but especially for cognitive and psychological experiments, is the timing and circumstance of testing. For instance, the attention paid by a participant to an online task can greatly modify the timing of responses, such as reaction time. Given that participants were given unlimited time to respond, and that reaction times reported in the study were in the ~1250ms (>1sec) range might perhaps suggest that people may not have been paying attention to the task. To me, this is a fatal error in experimental design because it directly confounds the main outcome measure in this study. Experimentally, one way to circumvent this problem is to introduce catch trials or trials with attention probes, which can help determine if participants were in fact paying attention to the task at hand. How did the authors deal with this issue? It is not clear from the Methods if online data collection was remote, or in-person? This should be specified along with detailed description of instructions etc. Was there supervision by an examiner, such as a proctor, or did people do this at their own leisure?

Along these lines, the finding that RT variability is significantly smaller in the autonomy group might indicate that participants in this group (owing to the need to make choices at every trial) may have become better at paying attention to the task—perhaps something of interest as it shows a different route through which such active decision making might help make performance more consistent and less variable.

On a related note, the alternate task switching was described as “difficult” and likened elsewhere in the Discussion as being “complex.” This may very well be true. However, as described above, a control condition is required to infer that. Indeed, a rather straightforward control condition would have been to have all participants perform a short block of simple reaction time task for the same keys. This would not only provide a baseline for comparison with the alternate-task switching task but could also be used to infer regarding attention (an issue discussed in the previous paragraph). In other words, if SRT>CRT (in the alternate task) then one could be more certain that the experimental alternate-switching task was not only more difficult but warranted more attention by participants (who exhibited longer RTs) thus suggesting that they were in fact paying more attention to this task.

However, there is no mention that any of the above or similar control measures were taken.

Validity of the findings

All in all, the lack of appropriate controls and sufficient methodological description (especially considering that online testing is a fairly new arena for psychological, psychomotor and cognitive fields), do not allow clear conclusions to be drawn from this study.

---

## Round 0.2 · accepted · Accept

I'd like to commend you and your colleagues for the manuscript itself and the revision you made on the basis of reviewers' comments, as well as to thank you for your patience during the review process.

Reviewer 1 ·

Basic reporting

I believe the authors have addressed all of my comments. The revised version of the manuscript is clear and well reasoned.

Experimental design

The added limitations and updated exploratory analyses are welcomed.

Validity of the findings

The study provides an unbiased estimate of the effect it sought to study and augments the extant literature.

Additional comments

I thank the authors for their revisions and for their patience in waiting for my review. I recommend accepting the manuscript.